# Metric Learning for Patch Classification in Digital Pathology

**Eu Wern Teh** ETEH@UOGUELPH.CA and **Graham W. Taylor** GWTAYLOR@UOGUELPH.CA
*School of Engineering, University of Guelph*
*Vector Institute for Artificial Intelligence*

## Abstract

We consider the problem of patch classification in digital pathology. We introduce a simple yet effective way to boost patch classification performance via metric learning. We hypothesize that the self perturbation and contrastive loss are both useful in improving generalization of the classification model. In our experiments with the PCam dataset (Veeling et al., 2018), we showed that models trained with both losses indeed outperformed our baseline where only cross-entropy loss is used. In addition, we also achieved state-of-the-art results on the PCam dataset (90.36% accuracy).

**Keywords:** Metric Learning, Digital Pathology, Patch Classification

## 1. Introduction

In digital pathology (DP), classifying gigapixel-sized whole slide images (WSI) is a very challenging task. Many off-the-shelf methods will not work directly with DP because resizing gigapixel images loses granular information, which is vital in differentiating between normal and tumor cells. Moreover, processing entire gigapixel images directly poses algorithmic and computational challenges. A commonly used pipeline first identifies tissue regions in WSI. This is followed by extracting patches that correspond to tissue and training a patch tumor classifier, which is then used to generate a heatmap for the WSI. Finally, another tumor classifier is trained using the morphological and geometrical information from the thresholded heatmap as features to classify WSI.

Current techniques (Liu et al., 2017; Lee and Paeng, 2018; Wang et al., 2016) for patch classification in DP optimize the cross-entropy loss. However, these techniques do not exploit comparison between data samples which we hypothesize is a useful way to regularize the network. In this work, we explore how this can be achieved through metric learning.

Metric Learning is a task to learn the similarity and dissimilarity between examples. It is commonly used for re-identifying objects such as people (Rahman et al., 2017), animals (Deb et al., 2018) or insects (Murali et al., 2019). In the context of classification, the benefit of metric learning over traditional setups (e.g. softmax) is that it does not assume the number of individuals to be fixed. Recently, metric learning has also been used in semi-supervised learning (Laine and Aila, 2016; Tarvainen and Valpola, 2017), which allows a model to be trained with little supervised information by utilizing unlabeled data. Conventionally, features are learned by training an auto-encoder on the unlabeled data where the difference between original image and reconstructed image is minimized (Rasmus et al., 2014). With metric learning, features can be learned by self-comparison under different perturbations.

**Contribution:** We use metric learning to improve the generalization of patch tumor classification models. Specifically, we learn two distance metrics: a) distance between perturbed samples and b) distance between samples from the same classes and different classes. We show that both distance metrics can be optimized together yielding a stacking improvement over one another. Our proposed models are illustrated in Fig. 1.

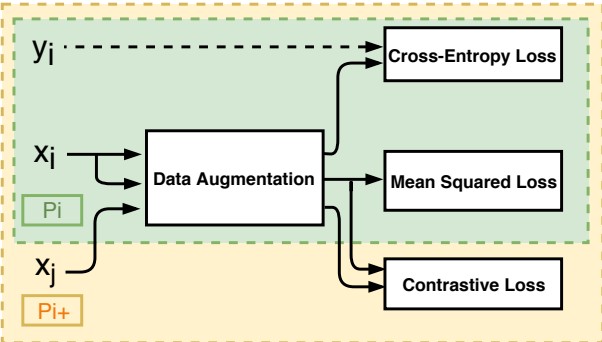

Figure 1: The proposed models: Pi and Pi+. The Pi model uses the self-perturbation loss which is the mean squared distance between the classification outputs of perturbed samples. The Pi+ model uses an additional loss function, contrastive loss which minimizes intra-class differences and maximizes inter-class differences.

## 2. Methods

Our method uses a standard CNN-based softmax classifier augmented with additional losses.

**Self-Perturbation Loss:** One way to learn the similarity between samples is by self-comparison. This can be achieved by minimizing the differences between the same sample under different perturbations (Laine and Aila, 2016). Let $x_i$ be a training example and $z_i$ and $z_i^*$ be the pre-softmax predictions of $x_i$ under different perturbations. Perturbation in our experiment is done via data augmentation which is described in Sec. 3. Self-Perturbation Loss, $L_1$ is defined as the mean squared differences between $z_i$ and $z_i^*$.

$$L_1 = \frac{1}{N} \sum_i^N (z_i - z_i^*)^2 \tag{1}$$

**Contrastive Loss:** Another way to learn the distance metric between samples is via contrastive learning. Koch et al. introduced the use of the contrastive loss in a Siamese Network for the one-shot learning task (Koch, 2015). Let $x_i$ and $x_j$ be two randomly chosen inputs with $y_i$ and $y_j$ labels and $z_i$ and $z_j$ the pre-softmax prediction of $x_i$ and $x_j$. The distance between $z_i$ and $z_j$ is defined as $\text{Dist}(i,j) = \frac{1}{N} \sum_i^N (z_i - z_j)^2$. The contrastive Loss, $L_2$ is defined as:

$$L_2 = \frac{1}{N} \sum_i \mathbb{1}(y_i = y_j) * \text{Dist}(i,j)) + \mathbb{1}(y_i \neq y_j) * \max(0, 1 - \text{Dist}(i,j)) \tag{2}$$

## 3. Experiments

**Dataset:** We experimented on PCam, a subset of the CAMELYON16 dataset (Bejnordi et al., 2017), which consists of CAMELYON16 image patches at $10\times$ magnification with a train/validate/test split of $74/12.5/12.5\%$. There are $327,680$ total patches of size $96 \times 96$ pixels. Similar to (Veeling et al., 2018), we trained our model on the training set and use the validation set for hyper parameter tuning. We reported our results on the test set.

**Experiment Settings:** ResNet-34 (He et al., 2016) architecture is used in all of the experiments. The models were trained using the Adam optimizer (Kingma and Ba, 2014) with an initial learning rate of $1e^{-4}$ for 10 epochs. The learning rate is reduced by a factor of 10 at the 6th epoch. During training, various data augmentations were performed, these include: a) Random Cropping b) Random Horizontal Flip c) Random Rotation ($0°$ to $360°$) and d) Color Jittering. In detail, we use Pytorch Deep learning framework in our experiments. We also use TorchVision package for data augmentation. Random cropping of size $224 \times 224$ were performed on padded images ($256 \times 256$), where the padded regions are generated via reflection. Color jittering was performed by setting the brightness, contrast, saturation and hue to 0.4. The best performing model on validation data is kepted at each epoch. During testing, images were resized to $224 \times 224$ without any augmentation.

| Method | Accuracy | Confidence Interval | Significance Test (p-value) |
|---|---|---|---|
| Baseline | $89.21 \pm 0.56$ | $(88.81, 89.61)$ | — |
| Pi | $\mathbf{89.94} \pm 0.34$ | $(89.69, 90.18)$ | 0.003 |
| Pi+ | $\mathbf{90.36} \pm 0.41$ | $(90.06, 90.65)$ | 0.024 |
| P4M(Veeling et al., 2018) | 89.80 | — | — |

Table 1: Accuracy of our models on PCam dataset. The experiment was run 10 times with different random seeds and we report the mean and standard deviation for each method. We also compared our models with P4M, which is the state-of-the-art architecture. In addition, we also we report 95% confidence interval for each method and perform significance testing for a) Baseline vs Pi (row 2) b) Pi vs Pi+ (row 3). The p-values indicate that the results are statistically significant.

**Results:** The results of the experiments are shown in Table 1, where the mean accuracy and standard deviation were reported. We see that the Pi model outperforms the baseline model. In addition, Pi+ model is better than Pi model and is able to outperform the state-of-the-art architecture (Veeling et al., 2018).

## 4. Conclusion

The experimental results showed that both distance metrics can be optimized together yielding a stacking improvement over one another. This work suggests that metric learning works in DP. Future work could focus on advanced metric learning methods, semi-supervised learning and image retrieval problems.

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
