# OpenReview forum: "Metric Learning for Patch Classification in Digital Pathology"
_MIDL.io/2019/Conference/Abstract — MIDL Abstract 2019_

### Official Review · AnonReviewer1 · 2019-04-30
**Improvement on state-of-the-art results using combination of metric learning**

**Rating:** 3
**Confidence:** 2

**Review:**


Summary:

A metric learning approach for patch classification using a combination of self-perturbation and contrastive losses is shown to be beneficial in digital pathology tasks. Experiments show significant improvement on state-of-the-art results.

Comments:

+ Mean-squared loss computed with data augmentation based perturbation to learn self-similarity between samples is a useful idea, and can be applied to other medical segmentation tasks as well.
+ Experiments are convincing and the paper is clearly written

- It is unclear as to which of the data augmentation steps were used as perturbation to learn the self-similarity.

---

### Official Review · AnonReviewer2 · 2019-04-30
**State-of-the-art results for DP patch classification**

**Rating:** 4
**Confidence:** 2

**Review:**

The authors present a CNN model using metric learning to classify digital pathology (DP) patches, which out-performs the state-of-the-art. The motivation for the self-perturbation and contrastive losses is made clear as potential regularizers for the case of DP patch classification.

A ResNet-34 model is trained and evaluated on the PCam dataset, with the baseline model using a standard cross-entropy (CE) loss, the Pi model using the CE+self-perturbation loss, and the Pi+ model using the CE+self-perturbation+contrastive losses.

The results demonstrate that the baseline model performs worse than the state-of-the-art model from (Veeling et al., 2018) (which uses an entirely different architecture), however the models with the proposed losses out-perform the state-of-the-art. Statistical significance is also assessed, demonstrating improvements over the baseline with both the Pi and Pi+ models.

This is a very clearly presented paper and nicely demonstrates the value of metric learning for a relevant medical imaging application.

---

### Decision · Program_Chairs · 2019-05-06
**Acceptance Decision**

Accept